# EFFICIENT-DLM: FROM AUTOREGRESSIVE TO DIFFUSION LANGUAGE MODELS, AND BEYOND IN SPEED

## ABSTRACT

The token-by-token decoding nature of autoregressive (AR) language models limits their generation throughput, especially in common memory-constrained scenarios. To address this, diffusion language models (dLMs) have emerged as a promising paradigm to enable parallel, non-autoregressive generation for higher throughput. However, existing dLMs have either failed to deliver faster speeds than AR models or have been restricted to small model scales due to high training costs, resulting in limited capability. To this end, we build on pretrained AR models and develop a training framework to convert them into dLMs that excel in speed. First, we introduce a continuous pretraining scheme with *a block-wise attention pattern* that remains causal across blocks while enabling bidirectional modeling within each block, which we find to better preserve pretrained models' abilities than the fully bidirectional modeling used in prior work such as Dream. Second, to mitigate the training–test gap in mask token distributions, we propose *a position-dependent token masking strategy* that assigns higher masking probabilities to later tokens. Leveraging this framework, we conduct extensive studies of dLMs' attention patterns, training dynamics, and other design choices, providing actionable insights into scalable AR-to-dLM conversion. We also deliver the Efficient-DLM model family, which outperforms state-of-the-art AR models and dLMs with better accuracy–throughput trade-offs, e.g., Efficient-DLM 4B achieves +1.88% higher accuracy with $4.63\times$ throughput compared to Dream 7B, and +7.79% accuracy with $1.82\times$ throughput compared to Qwen3 1.7B.

## 1 INTRODUCTION

The success of large language models (LLMs) has been largely driven by autoregressive (AR) modeling, where tokens are generated sequentially left to right. Despite strong benchmark performance, AR models are constrained by token-by-token decoding, which limits generation throughput, especially in memory-bounded scenarios (e.g., small batch sizes) where hardware utilization is low.

To overcome the sequential bottleneck of AR decoding, diffusion language models (dLMs) (He et al., 2022; Sahoo et al., 2024; Nie et al., 2025; Ye et al., 2025) have recently emerged as an alternative paradigm. By leveraging iterative denoising steps, dLMs enable parallel, non-autoregressive generation and hold promise for higher throughput. However, despite their conceptual appeal, most existing dLMs have not delivered faster speed than AR models in practice (Nie et al., 2025; Ye et al., 2025), due to the limited compatibility with key-value (KV) caching and the limited parallelism during decoding. Although pioneering works (Arriola et al., 2025; Sahoo et al., 2025b;a) demonstrate potential speed-up on small-scale models (e.g., 110M (Arriola et al., 2025)) with limited downstream accuracy, successful scaling of dLMs to larger model sizes has been restricted by prohibitive training costs. Thus, dLMs' promise as an efficient AR alternative remains unrealized.

Considering that AR models learn only left-to-right modeling, while dLMs learn all possible permutations (Xue et al., 2025), which is more difficult and requires longer training, this work leverages pretrained AR models for initialization and systematically explores how to continuously pretrain them into dLMs that excel in generation speed. The key insight of our study is that, with a proper training scheme, pretrained AR models can be converted into faster dLMs that perform parallel decoding using the KV cache at relatively low training cost (on the order of 10B tokens), and extended continuous training (on the order of 100B tokens) allows for more aggressive parallel generation.

We achieve this through a continuous pretraining scheme with a block-wise attention pattern (Arriola et al., 2025), which preserves causality across blocks while enabling bidirectional modeling within each block. Extensive studies on attention patterns show that this approach better maintains the weight distributions of AR models than the fully bidirectional training adopted in prior work (Nie et al., 2025; Ye et al., 2025), leading to notably higher accuracy. Second, we identify a gap between training-time uniform token masking and test-time confidence-based token sampling. To close this gap and improve downstream accuracy, we propose a position-dependent token masking strategy. This builds on the observation that dLMs retain a left-to-right generation tendency due to the autoregressive nature of language, and thus incorporates the prior that, as the input sentence becomes less corrupted (i.e., closer to completing denoising), more tokens should be masked near the end of each block. Third, leveraging this training framework, we analyze attention patterns, training dynamics, and other design choices for scalable AR-to-dLM conversion, and introduce the Efficient-DLM model family, which outperforms both AR and dLM baselines with improved accuracy–throughput trade-offs. For instance, Efficient-DLM 4B achieves +1.88%/+7.79% higher average accuracy with $4.63\times/1.82\times$ throughput compared to Dream 7B/Qwen3 1.7B, respectively.

We expect these findings to provide practical guidelines for realizing dLMs' promise of faster, more efficient generation, and to inspire new dLM paradigms. The key takeaways and insights from this work are summarized as follows:

**Takeaways for converting pretrained AR models into faster dLMs**

- Attention pattern is key to AR-to-dLM conversion: Continuous training with block attention better preserves pretrained AR models' abilities than fully bidirectional modeling.
- Training block size matters: Too-small block sizes lack sufficient context for denoising, while too-large block sizes induce excessive corruption and weight changes. Training with proper block sizes can generalize well to other evaluation block sizes.
- Larger evaluation block sizes generally provide more opportunities for parallel decoding.
- Left-to-right generation tendency: dLLMs still exhibit this tendency during parallel generation, and mimicking it in training can boost generation quality.
- Training dynamics reveal scaling behaviors: Likelihood estimation improves steadily with training, allowing for more aggressive parallel decoding with marginal accuracy drop.
- Conditioning each corrupted block on clean context is essential for model accuracy.
- Preserving the token shift of AR models is unnecessary and even harmful.

## 2 Continuous Pretraining with Block-wise Attention

### 2.1 Our Training Scheme

Existing works that transform AR models into dLMs (Gong et al., 2025a; Ye et al., 2025) adopt fully bidirectional modeling, i.e., the entire sequence is randomly corrupted and all tokens are visible to each other, as shown in Fig. 1 (a) and (b). This training scheme suffers from the following drawbacks: (1) fully bidirectional attention increases the difficulty of applying KV caching; (2) the context is overly corrupted, particularly for later tokens, which increases training difficulty; (3) the fully bidirectional attention pattern diverges from the autoregressiveness of the AR initialization, resulting in larger weight drifts from the pretrained AR models.

In light of this, we propose continuous pretraining with a block-wise attention pattern, as shown in Fig. 1 (d). This scheme preserves causality across blocks while enabling bidirectional modeling within each block, following the attention pattern in (Arriola et al., 2025). It also ensures that each block is processed using clean context only, mimicking the block-wise decoding process at test time where all previous blocks are fully completed without mask tokens. This is achieved by concatenating the noisy tokens and the clean tokens as dLM inputs and applying the special attention mask shown in Fig. 1(d). Such a block-wise training scheme allows for seamless use of KV cache for improved efficiency, constrains the corruption within each block to maintain cleaner context, and preserves block-wise autoregressiveness, thereby mitigating weight drift and better inheriting the capabilities of the original AR model.

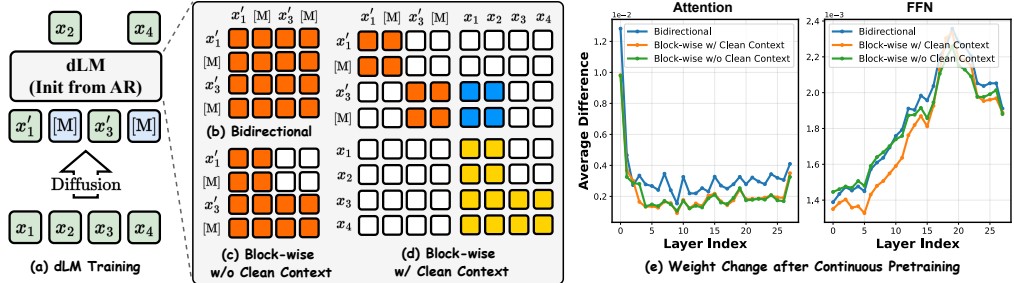

Figure 1: Visualizing continuous pretraining of dLMs with different attention patterns from pretrained AR models. (b) and (c) show bidirectional attention and block-wise attention without clean context, respectively, using a block size of 2 as an example. (d) illustrates our block-wise attention with clean context (using a block size of 2 as an example), where ■ denotes attention among noisy tokens, ■ denotes attention from noisy tokens to clean-context tokens, and ■ denotes attention within the clean context. (e) shows weight changes in the attention and feed-forward network (FFN) layers after continuous pretraining under these three attention patterns.

More formally, let $\mathbf{x} = (x_1, \ldots, x_L)$ be a sequence partitioned into $B$ contiguous blocks $\mathbf{x}^b$ of length $L' = L/B$, and let $q(\tilde{\mathbf{x}}_t^b \mid \mathbf{x}^b)$ denote the corruption process that produces the noisy input $\tilde{\mathbf{x}}_t^b$ at noise level $t \in (0, 1]$. We preserve causality across blocks while applying bidirectional denoising within each block. Specifically, the training objective is defined as follows:

$$\mathcal{L}(\theta) = \mathbb{E}_{t \sim \mathcal{U}[0,1]} \, \mathbb{E}_{\tilde{\mathbf{x}}_t^b \sim q(\cdot \mid \mathbf{x}^b)} \left[ -\frac{1}{t} \sum_{b=1}^{B} \log p_\theta(\mathbf{x}^b \mid \tilde{\mathbf{x}}_t^b, \mathbf{x}^{<b}) \right]. \quad (1)$$

where $p_\theta(\mathbf{x}^b \mid \tilde{\mathbf{x}}_t^b, \mathbf{x}^{<b})$ denotes the denoising of the $b$-th block based on the corrupted input $\tilde{\mathbf{x}}_t^b$ and the clean context $\mathbf{x}^{<b}$. We also compare with a block-wise attention variant that does not use clean context when processing a block: it replaces the clean context $\mathbf{x}^{<b}$ in Eq. 1 with the corrupted context $\tilde{\mathbf{x}}^{<b}$ when denoising the $b$-th block, as visualized in Fig. 1 (c).

Unlike prior block diffusion methods trained from scratch (Arriola et al., 2025), we initialize $\theta$ from a pretrained AR model that is trained with the autoregressive loss $\mathcal{L}_{\text{AR}}(\theta) = -\sum_{\ell=1}^{L} \log p_\theta(x_\ell \mid x_{<\ell})$, and then adapt it through continuous pretraining using the loss in Eq. 1. This initialization allows for rapid AR-to-dLM conversion, which requires (1) adapting weights to new attention patterns and (2) avoiding large weight drifts to better preserve the original model's ability.

## 2.2 EXPLORATION OF DIFFERENT TRAINING SCHEMES

We study the impact of three key design factors for AR-to-dLM conversion: (1) attention pattern: fully bidirectional vs. block-wise; (2) whether to keep clean context: using clean context $\mathbf{x}^{<b}$ or corrupted context $\tilde{\mathbf{x}}^{<b}$ in Eq. 1; and (3) whether to perform token shift, i.e., predicting the next token as in AR models or directly predicting the mask tokens themselves. Previous works (Gong et al., 2025a; Ye et al., 2025) find that preserving token shift when initializing from AR models is beneficial, and we revisit this design choice under more advanced training schemes.

**Settings.** We adopt Qwen2.5 1.5B (Team, 2024) as the AR initialization and perform continuous pretraining for 50B tokens on a mixed dataset comprising (Nano, 2025; Zhou et al., 2025; Fujii et al., 2025). For block-wise training, we adopt a block size of 16, and provide further analysis on block sizes in Sec. 2.3. The initial learning rate is set to 1e-5 and decayed to 3e-6 using a cosine schedule with the AdamW optimizer. An analysis of the learning rate is provided in Appendix D. We evaluate downstream task accuracy on six generation tasks, including HumanEval, HumanEval Plus, MBPP, MBPP Plus, GSM8K, and Minerva Math, using lm-evaluation-harness (Gao et al., 2024).

**Attention patterns.** As shown in Tab. 1, compared to bidirectional attention in Row (c), block-wise attention (even without clean context) in Row (d) can boost average accuracy by 8.94%. When combined with other best practices, i.e., conditioning on clean context and removing token shift in Row (g), block-wise attention improves the average accuracy over bidirectional attention by 19.12%. This implies that block-wise attention better preserves block-wise autoregressiveness and thus maintains the pretrained AR model's abilities more effectively than bidirectional attention, in addition to the benefit of native KV caching. Furthermore, visualizations of weight changes after continuous pre-

Table 1: Comparing different dLM training schemes on Qwen2.5 1.5B. Row (a) shows the accuracy of the original Qwen2.5 1.5B. Row (b) presents the training scheme of Dream (Ye et al., 2025). Row (g) shows the identified best scheme with block-wise attention, clean context, and no token shift.

| Row ID | Attn Pattern | Clean Context | Token Shift | KV Cache | Human -Eval | Human -Eval Plus | MBPP | MBPP Plus | GSM8K | Minerva Math | Avg |
|---|---|---|---|---|---|---|---|---|---|---|---|
| a | AR | - | ✔ | ✔ | 36.59 | 29.88 | 43.6 | 59.52 | 54.74 | 26.40 | 41.79 |
| b | Bidirectional | - | ✔ | ✘ | 15.85 | 12.20 | 16.2 | 24.34 | 28.96 | 11.08 | 18.10 |
| c | Bidirectional | - | ✘ | ✘ | 19.51 | 15.24 | 17.2 | 24.34 | 28.20 | 11.22 | 19.29 |
| d | Block-wise | ✘ | ✔ | ✔ | 31.10 | 25.61 | 23.6 | 36.77 | 38.44 | 13.88 | 28.23 |
| e | Block-wise (2×) | ✘ | ✔ | ✔ | 26.22 | 22.56 | 26.0 | 42.33 | 36.69 | 12.56 | 27.73 |
| f | Block-wise | ✔ | ✔ | ✔ | 38.41 | 33.54 | 33.0 | 48.68 | 51.48 | 21.04 | 37.69 |
| g | Block-wise | ✔ | ✘ | ✔ | 39.02 | 34.76 | 34.0 | 48.15 | 52.99 | 21.56 | 38.41 |

training in Fig. 1 (e) show that bidirectional attention leads to larger weight drifts from pretrained weights in both the attention and FFN layers, ultimately causing larger accuracy drops.

**The impact of clean context.** Based on the comparison between Rows (d) and (f) in Tab. 1, conditioning each block on clean context (Fig. 1 (d)) during training is critical, yielding a 9.46% accuracy improvement over using noisy context (Fig. 1 (c)). This is because, at inference time, the context preceding a noisy block has already been decoded without mask tokens; training with clean context therefore more closely mimics inference-time behavior. In addition, we train the noisy-context case in Fig. 1 (c) with a doubled token budget in Row (e) to account for the increased sequence length caused by concatenating noisy and clean tokens in Fig. 1 (d). However, comparing Rows (e) and (f) in Tab. 1 shows that doubling training tokens on corrupted context cannot effectively recover the accuracy, whereas training on fewer tokens with clean context yields substantially higher accuracy. Furthermore, training with block-wise attention without clean context leads to larger weight drifts in the early FFN layers compared to training with clean context, as shown in Fig. 1 (e).

**Whether to perform token shift.** We find that token shift is unnecessary, and its removal consistently improves accuracy across settings, as evidenced by the comparison between Rows (b) & (c) and Rows (f) & (g) in Tab. 1. This indicates that (1) the token shift inherent to AR models can be easily adapted into the no-token-shift setting, and (2) predicting the mask token itself (without token shift) is easier than predicting the next token of a masked position. We hypothesize that the latter is harder because the model must handle two tasks simultaneously: inferring the mask token and then predicting the following token.

**Takeaways.** When continuously pretraining from an AR model, a block-wise attention pattern with clean context and without token shift emerges as a promising training scheme to deliver dLMs. We adopt this scheme by default in the following study.

## 2.3 ANALYSIS OF THE OPTIMAL BLOCK SIZES

Building on the best block-wise training scheme in Sec. 2.2, the next question is the optimal block size for training and evaluation. Intuitively, larger context sizes provide richer context with more visible future tokens, but at the same time introduce more corruption, i.e., the last tokens in a block encounter noisier past context. This makes it critical to select a proper block size that balances both aspects. We study the impact of training and evaluation block sizes in this subsection.

**Settings.** We perform continuous pretraining on top of Qwen2.5 1.5B (Team, 2024) and Qwen3 4B (Yang et al., 2025) for 50B and 25B tokens, respectively, using different training block sizes [4, 8, 16, 32, 64, 128]. Other configurations are the same as in Sec. 2.2. We then evaluate each trained model with different evaluation block sizes and report the average accuracy across six generation tasks (HumanEval, HumanEval Plus, MBPP, MBPP

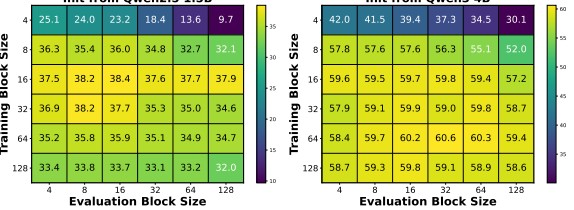

Figure 2: The average accuracy achieved by different training–evaluation block size pairs.

Plus, GSM8K, and Minerva Math) for each training–evaluation block size pair in Fig. 2.

**Observations on training block sizes.** As shown in Fig. 2, we observe that (1) For both model scales, too-small training block sizes generally lead to suboptimal accuracy, since the context is

not sufficiently rich to predict the corruptions. (2) Larger-scale models are more tolerant of larger training block sizes, which introduce more corruptions but also provide richer context. In contrast, small-scale models have a more notable sweet-spot training block size (e.g., 16 for Qwen2.5 1.5B), beyond which the more corrupted context leads to degraded accuracy. (3) Training with an appropriate block size can transfer well to other evaluation block sizes. This is consistent with the attention map in Fig. 1 (d), where the model trained with a single block size can see varying numbers of tokens participating in the attention mechanism. This differs from the fully bidirectional attention in Fig. 1 (b), where the model always sees the same number of tokens participating in attention, necessitating additional techniques such as random sequence length truncation (Nie et al., 2025) to generalize to other sequence lengths.

We also visualize the weight changes before and after continuous pretraining with different block sizes on top of Qwen2.5 1.5B in Fig. 3. In general, larger training block sizes lead to larger weight changes, and there exists a sweet-spot block size that yields the best downstream task accuracy. This indicates a trade-off between maintaining original abilities and adapting to new attention patterns, and the sweet-spot block size balances both aspects.

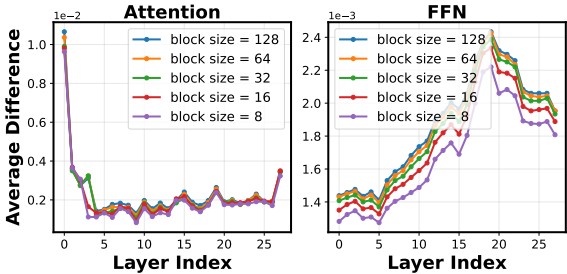

Figure 3: Visualizing the weight changes in attention and FFN layers after training with different block sizes.

**Observations on evaluation block sizes.** To understand the impact of evaluation block sizes for a trained model, we adopt confidence-based sampling (Wu et al., 2025) with different confidence thresholds to control the number of function evaluations (NFEs) (Ou et al., 2024). The lower the NFE, the more tokens are generated in parallel. We adopt diffusion Qwen2.5 1.5B and Qwen3 4B, trained with the best block sizes from Fig. 2 (16 and 64, respectively), and evaluate them with different block sizes. As shown in Fig. 4, we observe that (1) larger

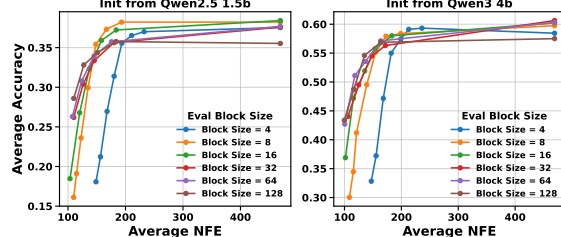

Figure 4: The average accuracy on six generation tasks of different evaluation block sizes under varying NFEs.

evaluation block sizes generally lead to higher accuracy when performing more aggressive token generation with lower NFEs. We assume this is because, under the same number of denoised tokens per step, larger block sizes provide more flexible positions and greater opportunities for parallel token generation; and (2) with larger NFEs, there is no clearly optimal evaluation block size, and moderate block sizes generally yield comparable results.

**Takeaways.** There exists a sweet-spot training block size: too-small block sizes lack sufficient context for denoising, while too-large block sizes induce excessive corruption and weight changes. In addition, although a proper training block size can generalize well to other evaluation block sizes, relatively larger evaluation block sizes favor more aggressive parallel token generation. In general, a block size of 32 or 64 is suitable for both training and evaluation of larger dLMs.

## 3 POSITION-DEPENDENT TOKEN MASKING

### 3.1 THE TRAINING-TEST GAP IN TOKEN MASKING

Existing dLMs (Nie et al., 2025; Ye et al., 2025; Arriola et al., 2025) typically adopt uniform token masking, where mask tokens are randomly sampled from a uniform distribution based only on the noise level $t$, independent of token positions. However, we find that at inference time, when performing confidence-based sampling (Nie et al., 2025; Ye et al., 2025), the denoised tokens are not uniformly distributed; instead, they show a clear left-to-right tendency.

To demonstrate this, we visualize the average number of denoising steps required at each token position in a block on the GSM8K dataset using the trained diffusion Qwen2.5 1.5B model from Sec. 2.2 in Fig. 5 (a). Specifically, we visualize two cases, with and without parallel token generation, using a confidence threshold (Wu et al., 2025), where tokens with confidence surpassing this threshold are

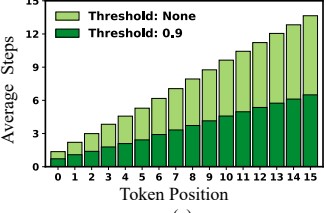 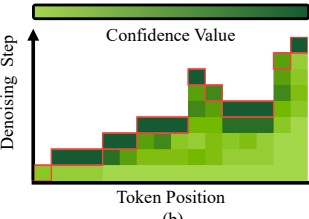 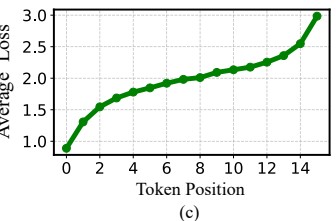

Figure 5: (a) The average number of denoising steps required at each token position on GSM8K using diffusion Qwen2.5 1.5B with two different confidence thresholds for parallel decoding, where "None" denotes one token per step. (b) The confidence distribution within a block across different denoising steps for an example from GSM8K, with red boxes marking tokens that are decoded and finalized. (c) The average loss at each token position within a block of size 16.

decoded. We observe that the average number of denoising steps increases with the positions in a block, exhibiting a notable left-to-right tendency due to the autoregressive nature of language.

As a concrete example, we also show the confidence distribution within a block across different denoising steps for one example from GSM8K in Fig. 5 (b), with red boxes marking tokens that are decoded and finalized. We can see that tokens tend to have higher confidence scores once their neighboring tokens have been decoded and are generally decoded from left to right. In other words, as the denoising process approaches completion, mask tokens are more likely to appear near the block end. As such, uniform token masking during training and confidence-based sampling during inference create a training–test gap.

In addition, we visualize the average loss at each token position within a block in Fig. 5 (c), averaged over all blocks for 200 samples. We observe that the later mask tokens in a block are generally harder cases with larger losses due to more corrupted context, potentially requiring more learning. This also indicates that a more strategic token masking scheme that also considers token position is desirable.

### 3.2 FORMULATION OF POSITION-DEPENDENT TOKEN MASKING

To demonstrate that the identified training–test gap and token masking scheme are important design factors, we propose the concept of position-dependent token masking. Specifically, for a sequence $\mathbf{x} = (x_1, \ldots, x_L)$ and a given noise level $t$, conditioned on $t$ and the relative token position $i \in [L']$ within one block, the masking probability of each token position is set as

$$w_i(t) = \exp\big[\beta\,(1-t)\,i\big], \tag{2}$$

where $\beta \geq 0$ is a hyperparameter controlling the strength of the positional bias. Specifically, $\beta = 0$ leads to uniform sampling, and larger $\beta$ indicates a stronger positional bias. The set of mask tokens is drawn from this distribution by normalizing the weights and then performing Gumbel-top-$k$ sampling (Huijben et al., 2022), where $k = \lfloor tL' \rfloor$ is the per-block mask token count.

When $t \to 0$, corresponding to the end of denoising, $w_i(t)$ assigns larger weights to later tokens, i.e., mask tokens are more likely to appear near the block end, echoing the test-time pattern in Sec. 3.1. When $t \to 1$, corresponding to noisier inputs, $w_i(t)$ becomes more uniform, i.e., mask tokens are sampled more uniformly. Such a position-dependent token masking narrows the training–test gap and increases the masking probability of later tokens, i.e., the harder cases with larger losses. We also note that related work (Wu et al., 2023) leverages the AR nature of language and designs a left-to-right noise schedule for continuous dLMs, whereas our work focuses on discrete dLMs.

### 3.3 COMPARISON OF TOKEN MASKING SCHEMES

**Settings.** We apply position-dependent token masking with different $\beta$ to the training of diffusion Qwen3 4B with block size 64 on 25B tokens. In practice, instead of directly setting $\beta$, we parameterize the positional prior using a half-life ratio $\lambda = \ln 2/(\beta L') \in (0, 1]$. The half-life ratio $\lambda$ is the fraction of a block length over which, under maximal tilt $t \to 0$, the positional weight changes by a factor of two. Thus, the lower the value of $\lambda$, the stronger the positional prior.

We compare position-dependent token masking with different $\lambda$ values against uniform token masking (i.e., $\lambda \to \infty$) and right-to-left masking (i.e., $\lambda \to 0$), which always masks the rightmost $k$ tokens. The average masking probability of each position within a block throughout training is

| Setting | TPF=1 | TPF=2.8 | TPF=4 | TPF=5.6 | Avg Diff |
|---------|-------|---------|-------|---------|----------|
| uniform | 60.27 | 57.12 | 51.11 | 33.99 | - |
| right-to-left | 38.21 | 27.55 | 18.60 | 14.13 | - |
| $\lambda$=0.25 | 60.41 (+0.14) | 58.56 (+1.44) | 50.97 (-0.14) | 34.55 (+0.56) | +0.50 |
| $\lambda$=0.1 | 62.02 (+1.75) | 58.79 (+1.67) | 53.75 (+2.64) | 38.37 (+4.38) | +2.61 |
| $\lambda$=0.05 | 60.51 (+0.24) | 57.68 (+0.56) | 53.06 (+1.95) | 37.38 (+3.39) | +1.54 |

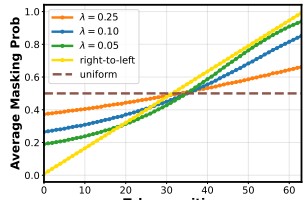

Table 2: Comparing token masking schemes based on the average accuracy across six generation tasks under varying parallel decoding settings, measured in tokens per forward (TPF).

Figure 6: The average masking probability of each token position within a block.

shown in Fig. 6. The average accuracy on the six generation tasks in Sec. 2.3, under different parallel decoding settings in terms of tokens per forward (TPF), i.e., the number of decoded tokens per denoising step using confidence-based sampling (Wu et al., 2025), is presented in Tab. 2.

**Observations.** As shown in Tab. 2, we observe that (1) progressively increasing positional priors with lower $\lambda$ leads to improved average accuracy; (2) positional priors are particularly beneficial under more aggressive parallel decoding settings, with up to a 4.38% average accuracy improvement; and (3) positional priors should not be blindly increased, as the extreme case of fully right-to-left masking without randomness leads to poor results, likely because the model is forced to train only on the hard cases at the block end without learning to exploit the bidirectional context.

These experiments indicate that positional priors are helpful but must be introduced properly. The key contribution of our work is to highlight this design factor, and we hope it can inspire more advanced and automated schemes in the future.

**Takeaways.** dLLMs exhibit a left-to-right tendency during parallel generation due to the autoregressive nature of language, and mimicking this tendency in training can boost generation quality.

## 4 ANALYSIS OF TRAINING DYNAMICS

dLM training with the objective in Eq. 1 improves the masked denoising likelihood under noisy conditions, but how this improved likelihood estimation translates into downstream task accuracy and parallel token generation ability remains unclear. In this section, we study the training dynamics of dLMs by visualizing the evolution of their performance on different tasks during training.

**Setting.** We train Qwen2.5 1.5B for 200B tokens with the same setting as in Sec. 2.3, and evaluate on both generation and likelihood-based tasks, where accuracy is computed by estimating and selecting the largest likelihood among multiple choices. We visualize the accuracy evolution on likelihood tasks and the accuracy–efficiency (NFE) trade-off across different training token budgets in Fig. 7.

**Observations and analysis.** We observe that (1) with relatively low training cost (on the order of 10B tokens), dLMs converted from pretrained AR models can largely recover task accuracy. (2) Longer training with more iterations consistently improves likelihood estimation and yields higher accuracy on likelihood-based tasks. The average accuracy on generation tasks, without considering parallel token generation (i.e., the rightmost points of each curve in Fig. 7 (b–d)), also improves, though with fluctuations on certain tasks. (3) Improved likelihood estimation allows for more aggressive parallel token generation, as reflected in the enhanced accuracy–NFE trade-off with longer training. This indicates that stronger likelihood estimation produces more accurate and reliable confidence scores, thereby improving generation quality under confidence-based sampling.

**Takeaways.** dLMs' ability to perform more aggressive parallel generation while maintaining accuracy improves with better likelihood estimation, which can be induced by longer training on more

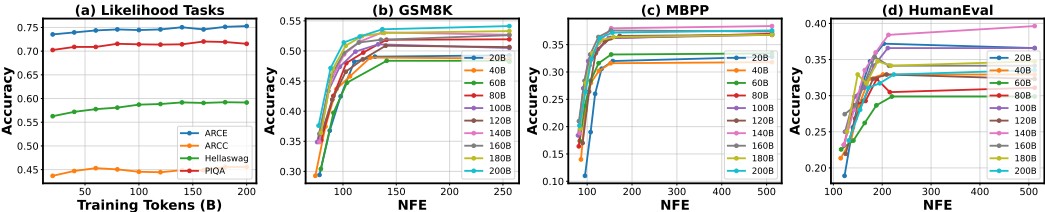

Figure 7: (a) The accuracy evolution on likelihood tasks during training. (b–d) The accuracy–NFE trade-offs across different tasks for models trained with varying token budgets.

Table 3: Benchmarking against SOTA AR models and dLMs on 12 tasks spanning coding, math, factual knowledge, and commonsense reasoning (CR), reporting average accuracy per category. TPF denotes tokens per forward, and TPS refers to throughput measured on an NVIDIA H100 GPU.

| Type | Model | TPF | TPS (tok/sec) | Coding | Math | MMLU | CR | Avg. |
|------|-------|-----|---------------|--------|------|------|-----|------|
| AR | Llama3.2 1B | 1.00 | 143.91 | 24.45 | 4.98 | 30.98 | 60.62 | 34.24 |
| | SmolLM2 1.7B | 1.00 | 112.84 | 21.06 | 30.97 | 49.99 | 68.44 | 40.14 |
| | Qwen2.5 0.5B | 1.00 | 99.93 | 31.90 | 25.97 | 47.65 | 55.31 | 41.98 |
| | Qwen2.5 1.5B | 1.00 | 73.03 | 42.17 | 46.98 | 60.96 | 66.00 | 54.47 |
| dLM | Efficient-DLM 1.5B | 2.33 | 158.89 | 42.33 | 42.60 | 57.63 | 62.58 | 52.04 |
| | | 2.69 | 184.48 | 41.79 | 41.80 | 57.63 | 62.58 | 51.77 |
| AR | Qwen3 1.7B | 1.00 | 71.59 | 54.22 | 54.15 | 62.53 | 64.99 | 59.39 |
| | Qwen3 4B | 1.00 | 47.13 | 63.85 | 66.27 | 73.19 | 70.91 | 67.97 |
| | Qwen3 8B | 1.00 | 42.51 | 68.45 | 69.87 | 76.93 | 73.71 | 71.58 |
| dLM | LLaDA 8B | 1.00 | 25.04 | 38.10 | 49.13 | 65.86 | 68.50 | 54.92 |
| | Dream 7B | 1.00 | 28.11 | 58.92 | 58.39 | 67.00 | 72.83 | 65.30 |
| dLM | Efficient-DLM 4B | 2.52 | 119.33 | 61.37 | 68.56 | 71.80 | 70.87 | 67.39 |
| | | 3.01 | 130.24 | 60.96 | 68.10 | 71.80 | 70.87 | 67.18 |
| dLM | Efficient-DLM 8B | 2.57 | 103.89 | 65.64 | 68.52 | 77.22 | 74.88 | 70.93 |
| | | 3.10 | 126.43 | 64.95 | 68.21 | 77.22 | 74.88 | 70.65 |

tokens. In addition, parallel token generation ability is another dimension for evaluating a dLM's performance: dLMs with comparable accuracy when denoising one token per step can exhibit notable accuracy gaps when performing aggressive parallel token generation.

## 5 EFFICIENT-DLM: A NEW FAMILY OF EFFICIENT dLMs

Combining previous insights, we develop the Efficient-DLM family with three sizes (1.5B/4B/8B), continuously pretrained from Qwen2.5-1.5B (block size 16) and Qwen3-4B/Qwen3-8B (block size 64), respectively. Our Efficient-DLM family integrates (1) the identified best attention pattern, i.e., block-wise attention with clean context and without token shift, as analyzed in Sec. 2, and (2) the position-dependent token masking with $\lambda = 0.1$ proposed in Sec. 3. Motivated by the training dynamics discussed in Sec. 4, we train for longer on 150B tokens using a mixed dataset comprising (Nano, 2025; Zhou et al., 2025; Fujii et al., 2025), adopting an initial learning rate of 1e-5 with cosine decay and the AdamW optimizer. Detailed settings are provided in Appendix B.

### 5.1 BENCHMARK WITH SOTA AR LMs AND dLMs

We benchmark our Efficient-DLM against SOTA AR LMs (Qwen3 (Yang et al., 2025), Qwen2.5 (Team, 2024), Llama3.2 (Grattafiori et al., 2024), SmolLM2 (Allal et al., 2025)) and SOTA dLMs (LLaDA (Nie et al., 2025) and Dream (Ye et al., 2025)) in Tab. 3. The benchmark covers 12 tasks, including math (GSM8k, Minerva Math), coding (HumanEval, HumanEval Plus, MBPP, MBPP Plus), factual knowledge (MMLU), and commonsense reasoning (ARCC, ARCE, Hellaswag, PIQA, Winogrande), as well as throughput measured on an NVIDIA H100 GPU with a batch size of 1. For each instance of Efficient-DLM, we report results under two parallel decoding settings with different tokens per forward (TPF). More detailed settings are provided in Appendix B.

**Observations.** As shown in Tab. 3, we observe that (1) compared to SOTA dLMs, our Efficient-DLM achieves both higher accuracy and efficiency. For example, Efficient-DLM 8B delivers 5.35% higher average accuracy with 4.50× throughput over Dream 7B, benefiting from the block-wise attention design over fully bidirectional modeling in continuous pretraining (see Sec. 2.2). (2) Compared to SOTA AR LMs, our Efficient-DLM attains better accuracy–throughput trade-offs. For instance, Efficient-DLM 8B/4B achieves 2.68×/1.82× throughput with +2.68%/+7.79% accuracy over Qwen3 4B/1.7B, respectively. More benchmarks with SOTA dLMs are in Appendix C.

### 5.2 ONE-FOR-ALL FLEXIBILITY: ADAPTIVE ACCURACY–EFFICIENCY TRADE-OFFS

Beyond efficiency, another advantage of dLMs is their one-for-all flexibility: a single dLM can balance accuracy and throughput to suit different deployment scenarios. This is achieved by controlling parallel tokens generation via a confidence threshold (Wu et al., 2025). Fig. 8 shows the one-for-all flexibility of our Efficient-DLM 4B across four tasks. Efficient-DLM maintains comparable accu-

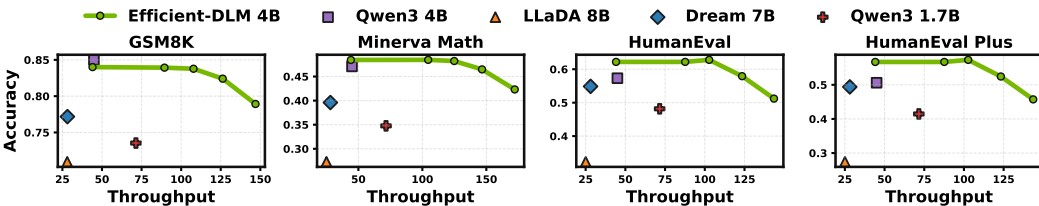

Figure 8: Visualizing the accuracy-throughput trade-off of different models on different tasks.

racy over a wide throughput range, e.g., a 4.96× speed-up on MBPP with a 1.8% accuracy loss. Throughput results under large batch sizes are provided in Appendix C.

## 5.3 Advantages of dLMs in Text Embedding

We further highlight that, thanks to the ability of bidirectional modeling, dLMs are more promising than AR models on tasks requiring bidirectional information. To demonstrate this, we evaluate our Efficient-DLM against AR Qwen models on text embedding tasks, benchmarking 15

Table 4: Comparing our Efficient-DLM and AR Qwen models on text embedding tasks (Muennighoff et al., 2022).

| Model | Retr. | Ranking | Clust. | Pair Class. | Class. | STS | Avg. |
|---|---|---|---|---|---|---|---|
| Qwen2.5 1.5B | 20.69 | 40.01 | 21.42 | 24.59 | 31.22 | 39.33 | 29.54 |
| Efficient-DLM 1.5B | 18.67 | 43.67 | 23.58 | 56.76 | 31.70 | 49.14 | 37.25 |
| Qwen3 4B | 19.46 | 39.90 | 21.77 | 33.94 | 29.13 | 40.56 | 30.79 |
| Efficient-DLM 4B | 20.17 | 45.05 | 23.91 | 65.59 | 42.22 | 47.27 | 40.70 |

datasets from the MTEB benchmark (Muennighoff et al., 2022) across six categories, following the ablation setup from LLM2Vec (BehnamGhader et al., 2024). As shown in Tab. 4, we observe a clear advantage of dLMs: at the 1.5B and 4B scales, Efficient-DLM outperforms AR Qwen models of the same sizes by 7.71% and 9.91% on average, respectively. These results also highlight the broader promise of dLMs for other sequence modeling tasks that require bidirectional information.

## 6 Related Work

**Diffusion language models.** To overcome the token-by-token decoding nature of AR LMs, diffusion LMs, both continuous (Li et al., 2022; Gong et al., 2022; Han et al., 2022) and discrete (Austin et al., 2021; He et al., 2022; Sahoo et al., 2024; Lou et al., 2023; Ou et al., 2024), have been proposed to perform non-AR decoding and thus enable parallel token generation. Among them, masked dLMs (He et al., 2022; Sahoo et al., 2024; Nie et al., 2025; Ye et al., 2025) have been successfully scaled up (e.g., LLaDA (Nie et al., 2025) and Dream (Ye et al., 2025)). Follow-up work has further explored alternative dLM paradigms (Sahoo et al., 2025b;a; Xue et al., 2025), and scaled them to larger generalists (Google DeepMind, 2025) or domain-specific specialists such as coding agents (Khanna et al., 2025; Song et al., 2025; Gong et al., 2025b; Xie et al., 2025).

**Diffusion language model acceleration.** Despite the acceleration potential of large dLMs (Nie et al., 2025; Ye et al., 2025), the gap between bidirectional attention and KV caching, along with the one-token-per-step denoising process, limits their achievable speed-up. To address these challenges, dedicated caching strategies for dLMs (Liu et al., 2025; Ma et al., 2025; Wu et al., 2025) have been developed to reuse computations and approximate bidirectional attention. In addition, to realize the potential of parallel token generation, confidence-based sampling (Wu et al., 2025), guidance from AR models (Israel et al., 2025), and adaptive decoding with certainty and positional priors (Wei et al., 2025) have been proposed. Beyond these training-free methods, Block Diffusion (Arriola et al., 2025) combines AR and diffusion by performing block-wise AR and in-block diffusion to support native KV caching. In parallel, to accelerate dLM training, (Gong et al., 2025a; Ye et al., 2025) propose initializing dLMs from AR models with token shifts.

## 7 Conclusion

This work systematically explores how to convert pretrained AR models into dLMs that achieve faster generation while retaining strong accuracy. By introducing a continuous pretraining scheme with a block-wise attention pattern, along with a position-dependent token masking strategy that narrows the training–test gap, we provide a principled framework for delivering dLMs with both strong accuracy and speed, resulting in the Efficient-DLM model family. Through comprehensive analyses of attention patterns, training dynamics, and other design choices, our findings offer actionable insights that we hope will guide the community toward building efficient and scalable dLMs.

ETHICS STATEMENT

The authors confirm compliance with the ICLR Code of Ethics. This work investigates improving the efficiency of language models by converting autoregressive models into diffusion language models with parallel decoding abilities, which may help reduce computational costs and environmental impact. The datasets used in this study are publicly available, as described in Sec. 5.

We acknowledge that the trained models may inherit biases from the pretraining data and could potentially be misused for harmful content generation. While our contributions focus on efficiency and accuracy trade-offs, we will emphasize the importance of responsible downstream use when releasing our models.

REPRODUCIBILITY STATEMENT

We have provided sufficient details to enable reproduction of our results. Specifically, the datasets, training procedures, and evaluation settings of our models are described in the first paragraph of Sec. 5, Sec. 5.1, and Appendix B. In addition, the experimental setups for each ablation study are elaborated in their respective sections in Sec. 2.2, Sec. 2.3, and Sec. 3.3.

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

## A  USE OF LLMs

In the development of this work, we used LLMs solely for revising the grammar of our manuscript. The research ideas, methodology, experiments, and analyses were conducted by the authors.

## B  DETAILED EXPERIMENTAL SETTINGS

**Training settings.** To train the Efficient-DLM model family, we perform continuous pretraining on top of AR Qwen2.5 1.5B and Qwen3 4B. Specifically, we follow the best practices from Sec. 2.2, Sec. 2.3, and Sec. 3.3, adopting training block sizes of 16 and 64, respectively, and setting $\lambda = 0.1$ for position-dependent token masking. Both models are trained on 150B tokens from a mixed dataset comprising (Nano, 2025; Zhou et al., 2025; Fujii et al., 2025), with an initial learning rate of 1e-5 under cosine decay and the AdamW optimizer. Training is conducted on 128 NVIDIA H100 GPUs and takes 2 days for Efficient-DLM 1.5B and 3.5 days for Efficient-DLM 4B.

**Evaluation settings.** For all evaluations of our Efficient-DLM in Sec. 5.1, we follow the best practice from Sec. 2.3 and use evaluation block sizes of 16 and 32 for our Efficient-DLM 1.5B and 4B, respectively. We use lm-evaluation-harness (Gao et al., 2024) to evaluate AR baselines (Qwen3 (Yang et al., 2025), Qwen2.5 (Team, 2024), Llama3.2 (Grattafiori et al., 2024), SmolLM2 (Allal et al., 2025)); for dLMs (LLaDA (Nie et al., 2025) and Dream (Ye et al., 2025)), we adopt their official evaluation code. We benchmark 12 tasks covering math (GSM8k, Minerva Math), coding (HumanEval, HumanEval Plus, MBPP, MBPP Plus), factual knowledge (MMLU), and commonsense reasoning (ARCC, ARCE, Hellaswag, PIQA, Winogrande). Following Dream (Ye et al., 2025), we use 8-shot, 4-shot, 0-shot, 0-shot, 3-shot, and 3-shot settings for GSM8k, Minerva Math, HumanEval, HumanEval Plus, MBPP, and MBPP Plus, respectively. The maximum number of generated tokens is set to 512 for all tasks, except GSM8k, which uses 256 as in (Ye et al., 2025).

**Parallel decoding settings.** Following (Wu et al., 2025), we set a confidence threshold and decode all tokens that exceed the threshold at each denoising step to enable parallel decoding. Tokens per forward (TPF) and generation throughput (tok/sec), reported in Tab. 3, are averaged over all six generation tasks.

**Text embedding evaluation settings.** Following the ablation setting of LLM2Vec (BehnamGhader et al., 2024), we evaluate the models on six categories of tasks from MTEB (Muennighoff et al., 2022), including retrieval (SciFact, ArguAna, NFCorpus), reranking (StackOverflowDupQuestions, SciDocsRR), clustering (BiorxivClusteringS2S, MedrxivClusteringS2S, TwentyNewsgroupsClustering), pair classification (SprintDuplicateQuestions), classification (Banking77Classification, EmotionClassification, MassiveIntentClassification), and semantic textual similarity (STS17, SICK-R, STSBenchmark). In total, the evaluation covers 15 datasets.

To obtain sequence embeddings, we apply mean pooling over the last layer's hidden states across all tokens. For Qwen models, we use a causal attention mask, as switching to a bidirectional mask consistently degraded performance. For our Efficient-DLM models, we instead employ a bidirectional attention mask. All experiments are conducted in a zero-shot setting, directly using the pretrained weights of Qwen and Efficient-DLM without any fine-tuning.

## C  MORE BENCHMARKS WITH SOTA AR LMs AND dLMs

**The accuracy–throughput trade-off with larger inference batch sizes.** As a complement to Sec. 5.2, we further visualize the trade-off between accuracy and throughput under different batch sizes for multiple models on the GSM8K dataset. As shown in Fig. 10, we observe that (1) our Efficient-DLM 4B consistently improves accuracy–efficiency trade-offs across batch sizes compared to both AR models and dLMs, e.g., +8.9% accuracy with comparable throughput ($1.04\times$) over Qwen3 1.7B even at a batch size of 16.; and (2) the efficiency benefits of dLMs are more pronounced at small batch sizes, which correspond to more memory-bounded scenarios. Nonetheless, the efficiency advantage and one-for-all flexibility remain evident at large batch sizes, enabled by parallel decoding.

**Benchmark with SOTA dLMs plus Fast-dLLM acceleration.** Existing public dLMs, LLaDA (Nie et al., 2025) and Dream (Ye et al., 2025), both equipped with fully bidirectional attention, can be accelerated by Fast-dLLM (Wu et al., 2025) through partial KV caching and parallel decoding. We benchmark our Efficient-DLM 4B against Dream and LLaDA enhanced with Fast-dLLM's dual cache and parallel decoding, using different confidence thresholds to control the accuracy–throughput trade-off. As shown in Fig. 9, our Efficient-DLM 4B consistently achieves a better accuracy–throughput trade-off than Dream and LLaDA on GSM8K, demonstrating that Efficient-DLM constitutes a stronger dLM family.

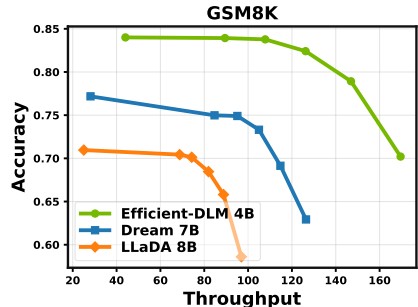

Figure 9: Comparing the accuracy-throughput trade-off with Dream/LLaDA plus Fast-dLLM.

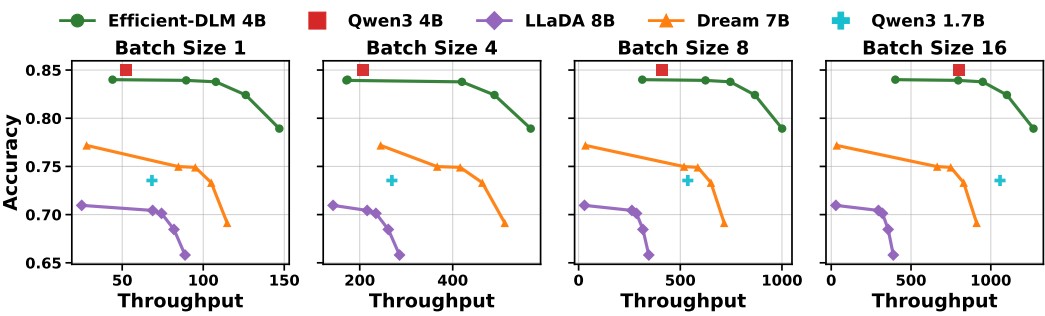

Figure 10: Comparing the trade-off between accuracy and throughput under different batch sizes for various models on the GSM8K dataset.

## D  THE IMPACT OF INITIAL LEARNING RATES

When initializing from pretrained AR models, the learning rate for continuous pretraining is a key hyperparameter, as it controls the speed of weight changes that affect both the preservation of the pretrained models' abilities and the adaptation to dLMs' new attention patterns. We perform an ablation study on Qwen3 4B trained for 25B tokens with different initial learning rates using a cosine learning rate schedule.

**Observations and analysis.** As shown in Tab. 5, we find that there exists a sweet-spot learning rate setting, e.g., 1e-5 in our case, that balances both aspects mentioned above. Intuitively, overly large learning rates cause greater weight drifts and degrade the pretrained models' original abilities, while overly small learning rates cannot effectively adapt to the new attention pattern. We also note that for any design factors in continuously training a pretrained model into dLMs, these two aspects should be carefully balanced to achieve decent final accuracy. Based on this set of experiments, we adopt 1e-5 as the default initial learning rate throughout the main manuscript.

Table 5: Comparing continuous training with different initial learning rates on Qwen3 4B.

| Init LR | HumanEval | HumanEval Plus | MBPP | MBPP Plus | GSM8K | Minerva Math | Avg |
|---------|-----------|----------------|------|-----------|-------|--------------|-----|
| 1.00E-04 | 49.39 | 43.90 | 44.20 | 56.08 | 72.56 | 39.54 | 50.95 |
| 3.00E-05 | 54.27 | 49.39 | 52.40 | 67.46 | 77.48 | 38.56 | 56.59 |
| 1.00E-05 | 57.93 | 51.22 | 54.40 | 71.96 | 81.73 | 46.54 | 60.63 |
| 3.00E-06 | 56.10 | 50.61 | 54.60 | 67.99 | 83.93 | 47.44 | 60.11 |
| 1.00E-06 | 45.73 | 42.68 | 47.2 | 66.67 | 81.12 | 43.94 | 54.56 |

## E  AR-TO-DLM CONVERSION VIA PARAMETER-EFFICIENT TUNING

Motivated by the relatively small weight changes observed in Sec. 2.2, we investigate whether parameter-efficient tuning can effectively convert pretrained AR models into dLMs. To this end, we apply Low-Rank Adaptation (LoRA) (Hu et al., 2022) to all linear layers in attention/FFN modules, combined with the best training scheme identified in Sec. 2.2, i.e., conditioning on clean context without token shift. All other parameters are frozen, except for the embedding layer, normalization operators, and the final model head, which we find must remain trainable for effective adaptation.

**Observations and analysis.** We extend Tab. 1 into Tab. 6 by including LoRA tuning results with two different ranks in Rows (h) and (i). We observe that LoRA tuning achieves reasonably good performance for AR-to-dLM conversion. Specifically, LoRA with rank 64 (Row i) surpasses the full-model training results of fully bidirectional attention and block-wise attention without clean context, while remaining 7.63% behind the full-model training results of the best scheme, i.e., block-wise attention with clean context. These results indicate that (1) with proper training schemes, even parameter-efficient tuning can yield competitive dLMs, and (2) full-model training remains necessary to obtain strong dLMs.

Table 6: Comparison of different dLM training schemes on Qwen2.5 1.5B. This table extends Tab. 1, with Rows (h) and (i) added to present the LoRA tuning results.

| Row ID | Attn Pattern | Clean Context | Token Shift | KV Cache | LoRA Rank | Human -Eval | Human -Eval Plus | MBPP | MBPP Plus | GSM8K | Minerva Math | Avg |
|---|---|---|---|---|---|---|---|---|---|---|---|---|
| a | AR | - | ✔ | ✔ | - | 36.59 | 29.88 | 43.6 | 59.52 | 54.74 | 26.40 | 41.79 |
| b | Bidirectional | - | ✔ | ✘ | - | 15.85 | 12.20 | 16.2 | 24.34 | 28.96 | 11.08 | 18.10 |
| c | Bidirectional | - | ✘ | ✘ | - | 19.51 | 15.24 | 17.2 | 24.34 | 28.20 | 11.22 | 19.29 |
| d | Block-wise | ✘ | ✔ | ✔ | - | 31.10 | 25.61 | 23.6 | 36.77 | 38.44 | 13.88 | 28.23 |
| e | Block-wise (2×) | ✘ | ✔ | ✔ | - | 26.22 | 22.56 | 26.0 | 42.33 | 36.69 | 12.56 | 27.73 |
| f | Block-wise | ✔ | ✔ | ✔ | - | 38.41 | 33.54 | 33.0 | 48.68 | 51.48 | 21.04 | 37.69 |
| g | Block-wise | ✔ | ✘ | ✔ | - | 39.02 | 34.76 | 34.0 | 48.15 | 52.99 | 21.56 | 38.41 |
| h | Block-wise (LoRA) | ✔ | ✘ | ✔ | 16 | 30.49 | 25.61 | 20.60 | 30.95 | 43.82 | 16.08 | 27.93 |
| i | Block-wise (LoRA) | ✔ | ✘ | ✔ | 64 | 28.66 | 25.61 | 24.40 | 40.21 | 48.14 | 17.64 | 30.78 |

## F  LOSS DISTRIBUTIONS ACROSS TOKEN POSITIONS

To study the difference in loss distributions across token positions between AR models and dLMs, we visualize the training loss defined in Eq. 1 for diffusion Qwen2.5 1.5B trained with a block size of 16, alongside the AR Qwen2.5 1.5B trained with the standard AR loss.

As shown in Fig. 11, which shows the average loss of the first 256 tokens in training sequences, we observe that (1) in AR models, the initial tokens incur higher loss due to the lack of context, while the loss of later tokens becomes more uniform; and (2) in dLMs, the loss follows a periodic pattern aligned with block boundaries, where later tokens within each block show higher loss due to limited clean context, consistent with Fig. 5 (c). In addition, similar to AR models, the initial tokens of the entire sequence also experience higher loss from insufficient context.

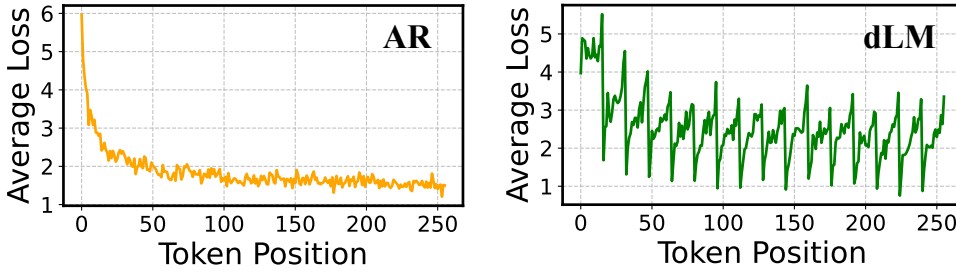

Figure 11: Visualizing the loss distributions over token positions of AR models and dLMs.

