# OpenReview forum: "Efficient-DLM: From Autoregressive to Diffusion Language Models, and Beyond in Speed"
_ICLR.cc/2026/Conference — Submitted to ICLR 2026_

### Official Review · Reviewer_at2L · 2025-10-27

**Soundness:** 2
**Presentation:** 3
**Contribution:** 2
**Rating:** 4
**Confidence:** 4

**Summary:**

This paper introduces Efficient-DLM, a Diffusion Language Model (dLMs) designed for high generation throughput and accuracy, achieved by continuously training pretrained Autoregressive (AR) models. The core contributions are a block-wise attention pattern that preserves AR model capabilities and enables KV caching, and a position-dependent token masking strategy to mitigate the training-test distribution gap. Efficient-DLM demonstrates superior accuracy-throughput trade-offs compared to state-of-the-art AR models and existing dLMs (LLaDA, Dream).

**Strengths:**

- The paper provides comprehensive empirical studies of the training recipes for discrete diffusion models, providing insights into factors that influence performance, efficiency and stability. These findings can serve as a practical reference for future research and development in discrete diffusion-based language modelling.
- The final model, Efficient-DLM, achieves better performance among a wide range of evaluations compared to other DLMs.

**Weaknesses:**

- My main concern is the actual contribution of this paper. To me, it appears to be a block-diffusion approach [1] that uses pretrained auto-regressive models to initialise the model weights. While the paper emphasises engineering discussion, the overall method seems to be a straightforward combination of previous ideas:
    - The block-diffusion concept is derived from [1].
    - The auto-regressive initialisation comes from [2].
    - The efficiency and acceleration result from leveraging KV-cache with block diffusion, as in [3].
- The paper reads more like an empirical study on the training of block-diffusion models. However, most of the observations are already well-known within the community, which limits the novelty and impact of the work.

[1] Arriola, Marianne, et al. "Block diffusion: Interpolating between autoregressive and diffusion language models." *arXiv preprint arXiv:2503.09573* (2025).

[2] Gong, Shansan, et al. "Scaling diffusion language models via adaptation from autoregressive models." *arXiv preprint arXiv:2410.17891* (2024).

[3] Liu, Zhiyuan, et al. "dllm-cache: Accelerating diffusion large language models with adaptive caching." *arXiv preprint arXiv:2506.06295* (2025).

**Questions:**

- It is encouraging to see that the proposed Efficient-DLM achieves the best average performance among discrete language models. However, it is unclear which specific components contribute most to this improvement. Adding more ablation studies, such as evaluating the effects of position-dependent masking, …, would help clarify the contributions of each component.
- It is surprising to see in Table 3 that Efficient-DLM achieves significantly better token-per-second (TPS) performance. Why is this the case? While block-diffusion is generally more efficient at test time due to the KV-cache, during training, I would expect it to be less efficient than standard discrete diffusion. This is because block-diffusion requires concatenating duplicated text to enable parallel training, which should typically reduce TPS.
- In Table 3, what is the difference between the two Efficient-DLM rows?

---

> ### Author Response · Authors · 2025-11-22
> **Response to Reviewer at2L (Part 1)**
>
> Thank you for acknowledging the comprehensiveness of our study and the performance of our delivered dLM, as well as for your constructive comments. We have addressed all of your questions below.
>
> ---
> **1. Contributions and novelty of our work**
>
> Thank you for pointing this out! We would like to humbly clarify our contributions and compare them with prior work as follows.
>
> Overall, our work is among the first (alongside some concurrent efforts) to study effective AR-to-dLM conversion that preserves the accuracy of pretrained AR models. Our work makes strong contributions by identifying the limitations in attention patterns and objectives (i.e., which tokens to mask and learn) in existing works, and by proposing the principles and methodology for successful AR-to-dLM conversion, which lead to a SOTA dLM family.
>
> ***(1)*** For attention patterns, our work identifies the correct attention pattern with a win–win in accuracy and efficiency through a comprehensive study, and simplifies unnecessary designs in prior works.
>
> **Compared to existing AR initialization works:** Although prior works like [2], as you mentioned, also studied AR-to-dLM conversion, their adopted bidirectional attention is a suboptimal design and leads to a significant accuracy drop according to Table 1 of their paper [2] and Table 1 of our manuscript. The key novel finding we provide is that *maintaining pretrained AR weight distributions with block-wise causality is critical for effective AR-to-dLM conversion*. This is an important insight we offer to the community and serves as a useful principle for future AR-to-dLM training techniques.
>
> In addition, [2] adopts token shifts (next-token prediction) to narrow the gap between AR and dLM, and we find that this extra design is unnecessary and simplify it.
>
> **Compared to the block diffusion work:** Although block diffusion ([1], as you mentioned) studied block-wise attention patterns, their primary aim is to enable caching. In contrast, we are the first to identify the benefits of block-wise attention in preserving causality for AR-to-dLM conversion, which leads to a +9% average accuracy improvement according to Table 1 of our manuscript. Thus, we actually propose a strong position for the community: *block-wise attention (with clean context) is the way to go, achieving a win–win in accuracy and efficiency for AR-to-dLM conversion*.
>
> Additionally, they conducted only small-scale experiments with per-block-size training/evaluation, whereas we scale up and deliver models that can adapt to different evaluation block sizes.
>
> ***(2)*** In addition to the contributions in identifying the correct attention patterns, we also propose a new position-dependent token masking scheme in Section 3 of our manuscript.
>
> We note that previous works study only the training–test gap in attention patterns of dLMs, while we are the first to pinpoint the training–test gap in token masking schemes and verify the effectiveness of more strategic token masking schemes. Beyond the proposed solution, this finding and the identified problem may be even more valuable in inspiring future AR-to-dLM training schemes.
>
> ***(3)*** We scale up our techniques to 1.5B/4B dLMs (we have also added 8B results, which are attached below) and deliver the Efficient-DLM model family that outperforms previous dLMs by a notable margin. More importantly, we also demonstrate the promise of dLMs in one-for-all flexibility for adaptive accuracy–efficiency trade-offs with a single model, as well as their benefits in stronger text embeddings. This is to inspire the community about other benefits of dLMs over AR models.
>
> Note that in addition to final model performance, we performed an extensive study on the impact of different design factors to shed light on the underlying principles (summarized in the key takeaways at the end of each section), which could serve as useful practical guidelines for future work.
>
> ***In summary:*** Although some components, such as block-wise attention, have been adopted by prior works (even before dLMs), our work discerns the correct direction to take (in terms of attention patterns) for AR-to-dLM conversion by analyzing the limitations of existing designs and conducting extensive studies to identify the underlying principles, as well as introducing novel techniques (position-dependent token masking). All together, these contributions lead to a SOTA dLM family.
>
>
> *New results for Efficient-DLM-8B:*
> | **Model** | **Token-Per-Forward** | **Throughput (tok/sec)** | **Coding (%)** | **Math (%)** | **MMLU (%)** | **CR (%)** | **Avg (%)** |
> |:---:|:---:|:---:|:---:|:---:|:---:|:---:|:---:|
> |Dream-7B|1.00|28.11|58.92|58.39|67.00|72.83|65.30|
> |LLaDA-8B|1.00|25.04|38.10|49.13|65.86|68.50|54.92|
> |Qwen3-8B|1.00| 42.51|68.45|69.84|76.93|73.71|71.58|
> |Efficient-DLM-8B (Ours)|1.00|40.00|67.36|69.22|77.22|74.88|71.62|
> |  | 2.57 | 103.89 | 65.64 |68.52|77.22|74.88|70.93|
> |  | 3.10 | 126.43 | 64.95 | 68.21 | 77.22 | 74.88 | 70.65 |

---

> ### Author Response · Authors · 2025-11-22
> **Response to Reviewer at2L (Part 2)**
>
> **2. The contribution of each component**
>
> Thank you for the good question! We believe it will be beneficial to highlight this in a single table, as the corresponding numbers and analyses are currently spread across different tables and figures in our submitted manuscript. We have organized the following table based on the reported results (for AR-to-dLM conversion on Qwen3 4B) in our manuscript to demonstrate the contribution of each component to the final accuracy:
>
> | Setting | HumanEval (%) | HumanEval Plus (%) | MBPP (%) | MBPP Plus (%) | GSM8K (%) | Minerva Math (%) | Avg (%) |
> |:---:|:---:|:---:|:---:|:---:|:---:|:---:|:---:|
> | Bidirectional attn (i.e., Dream's setting) | 39.02 | 32.32 | 39.60 | 50.00 | 67.40 | 39.17 | 44.59 |
> | Switch to Block-wise attn w/ clean context | 53.66 | 50.36 | 55.60 | 69.70 | 78.39 | 46.33 | 59.01 |
> | + Remove token shift | 56.10 | 51.22 | 54.60 | 69.84 | 82.87 | 47.02 | 60.27 |
> | + Position-dependent token masking | 60.37 | 54.27 | 59.00 | 71.43 | 81.12 | 45.92 | 62.02 |
> | + Scale to 150B tokens | 62.20 | 56.71 | 58.40 | 71.69 | 84.00 | 48.44 | 63.57 |
>
> *Note: Before scaling to 150B tokens, all experiments are trained for 25B tokens on top of Qwen3 4B.*
>
> As shown in the table above, attention patterns (with proper block-size selection), removing token shift, adding position-dependent token masking, and longer training all contribute to successful AR-to-dLM conversion.
>
> ---
> **3. Clarifications on the reported token-per-second (TPS) performance**
>
> We clarify that the reported token-per-second (TPS) refers to inference throughput, not training throughput. Since our dLMs can decode multiple tokens per forward pass, they naturally achieve higher inference throughput. We will further clarify this in the final version.
>
> Regarding training speed, as you mentioned, block-diffusion-style attention masks require more time per training step due to the doubled sequence length. To verify whether this doubled sequence length is worthwhile, we performed a comparison under an iso-compute setting in Rows (e) and (f) of Table 1 in our manuscript, comparing block-wise attention without clean context (with doubled training tokens) and block-wise attention with clean context. We find that clean context is critical: it leads to 10% higher average accuracy even when the training tokens are halved.
>
> ---
> **4. Difference between the two Efficient-DLM rows**
>
> As mentioned in Section 5.1 of our manuscript, we report two parallel decoding settings with different tokens per forward (TPF) at inference time for the same Efficient-DLM model. This is achieved by using different confidence thresholds, following fast-dLLM. A lower confidence threshold leads to more aggressive parallel token generation per model forward, resulting in higher throughput but lower accuracy.
>
> ---
> We believe that we have answered all your questions and concerns. Please let us know if you have any follow-up questions!

---

> > ### Author Response · Authors · 2025-11-26
> >
> > Dear Reviewer at2L,
> >
> > As we approach the end of the discussion period, we would greatly appreciate any further feedback you may have on our responses. We believe we have clarified and addressed all of your concerns and questions, and we welcome any additional comments or suggestions you might have. We are happy to continue the discussion.

---

### Official Review · Reviewer_mabx · 2025-11-02

**Soundness:** 3
**Presentation:** 3
**Contribution:** 3
**Rating:** 6
**Confidence:** 3

**Summary:**

This paper conducts conversion of AR models into diffusion LMs at scale, and offers several practical insights. Using these insights, paper introduces Efficient-DLM 4B, that achieves similar accuracy and higher throughput compared to AR baselines.

**Strengths:**

- conducts lots of experiments to provide plethora of takeaways for adapting autoregressive models to diffusion
- demonstrates speed ups compared to AR model with minimal accuracy drop
- due to the careful study, choosing the "right" block size can mean choosing varying block sizes at inference, showing accuracy-efficiency trade-offs

**Weaknesses:**

- the writing can be improved, right now the main paper is very dense and there are a lot of details to keep track of; when starting section 5, perhaps what changes are actually incorporated in the final efficient-DLM could be listed once again. Let me know what changes you plan to do to make the main paper more accessible and clear :)

**Questions:**

- how would the latency change when the context length increases?
- is there any KV caching within a block as well?

Since i'm not an expert in this field, i'll be looking at feedback of other reviewers too to gain a better understanding.
That being said, i'm definitely willing to improve my score during our discussion :)

---

> ### Author Response · Authors · 2025-11-22
> **Response to Reviewer mabx**
>
> Thank you for acknowledging the “plethora of takeaways” provided by our work and for your constructive comments. We have addressed all of your questions below.
>
> ---
> **1. Improvements in writing**
>
> Thank you for pointing this out! Our current writing follows the logic that we studied the attention patterns in Section 2, the training objectives (i.e., which tokens to mask and learn) in Section 3, the training dynamics when scaled up in Section 4, and the final delivered models when combining everything in Section 5. All studies were performed progressively, and we provided key takeaways after each section to summarize the findings.
>
> To make our manuscript more accessible, we plan to clarify the above structure with the following plan (we did not majorly revise the draft during the rebuttal phase since we want to keep the original sections for other reviewers’ reference, but we promise to revise it in the final version):
>
> - *Reorganize the sections*: We plan to add an overview after the introduction section, highlighting that the keys to AR-to-dLM conversion are (1) attention patterns and (2) training objectives (i.e., which tokens to mask and learn), which are elaborated in the current Sections 2 and 3, respectively. In addition, for the current Section 4 (the study on training dynamics), we plan to move it into a subsection of Section 5 (the delivered models) as an analysis of our delivered models. In this way, the overall structure becomes exploring the two key design components and then combining them to produce the best-performing dLMs.
>
>
> - *More structured observations*: For the rich observations in our comprehensive study, we plan to present them in a more structured way, e.g., using bullet points that follow the format of “observation + quantitative support.”
>
>
> - *Clarify the settings and transitions across sections*: When entering a new section, we plan to clarify which design choices are inherited from previous explorations and what is new to explore in the current section. (In fact, we currently always adopt the best setting from the previous section.)
>
> In addition, following your suggestion, we have highlighted the key changes and techniques included in the final delivered Efficient-DLM (highlighted in blue) at the beginning of Section 5 in the revised draft.
> Please let us know if you have other concrete suggestions for improving the writing. We would be happy to discuss them.
>
> ---
> **2. The throughput comparison under different context lengths**
>
> The reported throughput in our manuscript is based on the average context length in the evaluated math/coding tasks, which is on the order of 1k. Following your suggestion, we have added an analysis of the throughput comparison between our Efficient-dLM-4B and AR Qwen3 4B under different input context lengths (1k–64k), as shown in the table below.
>
> | **Model** | **Token-per-forward** | **Avg acc (%)** | **ctx=1k** | **ctx=2k** | **ctx=4k** | **ctx=8k** | **ctx=16k** | **ctx=32k** | **ctx=64k** |
> |:---:|:---:|:---:|:---:|:---:|:---:|:---:|:---:|:---:|:---:|
> | Qwen3-4B | 1.00 | 67.97 | 51.60 | 51.79 | 51.52 | 49.41 | 34.98 | 21.64 | 12.22 |
> | EfficientDLM-4B | 3.06 | 66.95 | 136.03 | 135.91 | 135.43 | 127.51 | 88.71 | 55.75 | 31.59 |
> | Speedup | - | - | 2.64x | 2.62x | 2.63x | 2.58x | 2.54x | 2.58x | 2.58x |
>
> *Note: Throughput (tok/sec) is measured on an NVIDIA H100 GPU with a batch size of 1. The input context length is shown in the table, and the generation length is fixed at 256.*
>
> We can observe that across different context lengths, the throughput improvement achieved by our Efficient-dLM is consistent. This is because all these scenarios are still memory-bounded, where predicting more tokens per model forward increases computation intensity with almost consistent cost per model forward.
>
> ---
> **3. Whether KV caching is also applied within a block**
>
> No, KV caching is applied at the granularity of a block, i.e., past blocks are cached, and the current processing block is not cached until it is completed.
>
> We also want to note that in Figure 2 of our manuscript, we showed the good adaptivity of a trained dLM to different evaluation block sizes, indicating that we can switch to smaller block sizes at inference time for more fine-grained KV caching and compute reuse if this becomes important in the target deployment scenarios.
>
> ---
>
> We believe that we have answered all your questions. Please let us know if you have any follow-up questions!

---

### Official Review · Reviewer_KCLd · 2025-11-03

**Soundness:** 2
**Presentation:** 2
**Contribution:** 2
**Rating:** 2
**Confidence:** 5

**Summary:**

This paper presents a systematic study on converting pretrained autoregressive (AR) language models into efficient diffusion language models (dLMs) that enable parallel generation while preserving accuracy. The authors address two key challenges:

1. Incompatibility between AR initialization and standard bidirectional dLM training: Solved via a *block-wise attention pattern* that maintains causality across blocks while permitting bidirectional modeling within blocks.
2. Training-test gap in token masking distributions: Tackled through a novel *position-dependent token masking strategy* that assigns higher masking probabilities to later tokens.

Their analysis shows that dLMs retain a left-to-right generation tendency despite parallel capabilities and that appropriate attention patterns greatly reduce weight drift from pretrained models. The Efficient-DLM family achieves strong accuracy-throughput trade-offs—e.g., Efficient-DLM 4B attains +1.65% higher accuracy with 4.77× throughput over Dream 7B, and +7.56% accuracy with 1.87× throughput over Qwen3 1.7B. The work offers actionable guidance for AR-to-dLM conversion and meaningfully bridges theoretical parallelism with practical generation speedups, especially for memory-constrained deployments.

**Strengths:**

- **Good recipe for AR→dLM conversion.** Combines block-wise attention (causal across blocks, bidirectional within), clean context conditioning, and position-dependent masking $w_i(t) = \exp[\beta(1-t)i]$ with a nice half-life parameterization ($\lambda = \ln 2/(\beta L')$). The design choices make sense and address real training-inference gaps.

- **Very thorough experiments.** Covers attention ablations, block sizes, long training runs (200B tokens), LoRA-based conversion, plus detailed diagnostics (weight drift, position-wise loss). The accuracy-throughput analysis across coding/math/knowledge tasks is pretty comprehensive.

- **Paper is clear.** Eq. (1) is well-stated, figures show the attention patterns nicely, and the half-life semantics make the masking strategy easy to understand.

**Weaknesses:**

**Eq. (1) doesn't match the implementation description:**

Eq. (1) sums over all blocks ($b = 1..B$), but §2.1 and Fig. 1(d) say each term conditions on a clean prefix ($\mathbf{x}^{<b}$) plus a corrupted current block ($\tilde{\mathbf{x}}_t^b$). If every block is corrupted simultaneously, you can't keep all the prefixes clean unless you replicate the sequence $B$ times.

If training corrupts exactly one block per sample (keeping prefix clean), the objective should be:

$$
\mathbb{E}_{b \sim \mathrm{Unif}[B]}\mathbb{E}_{t,\tilde{\mathbf{x}}_t^b} \Big[-\frac{1}{\alpha_t} \log p_{\theta}(\mathbf{x}^b \mid \tilde{\mathbf{x}}_t^b,\mathbf{x}^{<b})\Big]
$$

not a sum over all blocks. As written, Eq. (1) suggests either (a) multiple forward passes per sequence (one per block), or (b) an impossible conditioning setup.

This also conflicts with Row (e) in Tab. 1 ("doubled token budget to account for the increased sequence length caused by concatenating noisy and clean tokens"). If the sum is truly over all $b$, compute is already multiplied; if selecting one block per sample, Eq. (1) shouldn't have a sum.

---

**Inconsistent mask count:** The paper defines per-block positional weights $w_i(t)$ for $i \in [1..L']$, but then says "The set of mask tokens is drawn … with $k = \lfloor t L \rfloor$." If sampling is per block, shouldn't $k$ be $\lfloor t L' \rfloor$ instead of $\lfloor t L \rfloor$? Using total sequence length ($L$) seems inconsistent with the per-block approach and would ask for more masks than fit in a block.

---

**Best dLM still lags behind the AR source.** In Tab. 1, Row (a) (AR Qwen2.5-1.5B) "Avg" is 41.79, but the best dLM setting (Row (g): block-wise + clean context + no shift) only hits 38.41 after 50B tokens. That's a −3.38 point drop (≈−8.1% relative). The text says block-wise attention "better preserves pretrained AR models' abilities"—which is true compared to *other* dLMs—but at the same model size the proposed approach doesn't recover the AR baseline accuracy.

---

**Missing definition:** $\alpha_t$ is used but never defined.

---

**Novelty concerns:** The core recipe (block‑wise attention with clean prefix + confidence‑based parallel decoding + KV caching) is quite close to Block Diffusion (which also does AR across blocks, diffusion within blocks, with native KV caching). The "continuous pretraining from AR" idea was originally in DiffuLLaMa, which isn't cited here.

**Questions:**

- The embedding evaluation doesn't include LLM2Vec (cited tho), which also does continuous training from pretrained AR models and supports bi‑mask + contrastive training, why?

Please also see above section.

---

> ### Author Response · Authors · 2025-11-22
> **Response to Reviewer KCLd (Part 1)**
>
> Thank you for acknowledging the thoroughness of our study, as well as for your constructive comments. We have addressed all of your concerns and questions below.
>
> ---
> **1. Eq. (1) exactly matches our implementation**
>
> We humbly clarify that **our implementation exactly matches Eq. (1)**, where we train on all corrupted blocks within one training step (one forward/backward) through the specialized attention pattern shown in Figure 1 (d) of our manuscript.
>
> Specifically, in each training step, we concatenate the noisy tokens (with all blocks corrupted simultaneously) and the clean tokens, as shown in the attention mask in Figure 1 (d). The yellow part of the attention pattern, which is block-wise attention among clean tokens, is used to compute the clean context at each layer. The noisy tokens in each block condition on the clean context from previous blocks (i.e., the blue part of the attention pattern) and on the noisy tokens in the current block (i.e., the orangered part of the attention pattern). This provides an efficient way to compute the loss across all corrupted blocks in Eq. (1) of our manuscript exactly within one forward/backward pass, implemented using FlexAttention.
>
> We also note that the cost of such an attention pattern is a doubled sequence length per sample. We compared this design with pure block-wise attention without conditioning on clean context but trained with doubled tokens in Rows (f) and (e) of Table 1 in our manuscript, respectively. We find that the former achieves +9.96% average accuracy, indicating that conditioning on clean context is worthwhile.
>
> Thank you for pointing out this confusion, and we have clarified this (highlighted in blue) in the revised draft.
>
> ---
> **2. Mask count per block**
>
> Thank you for pointing this out. You are right that the per-block mask token count is $k = \lfloor tL' \rfloor$, where $t$ is the noise level and $L'$ is the block length. Our intention was to state that the total mask token count in each sample is $k = \lfloor tL \rfloor$, with the per-block masking probability parameterized by Eq. (2) of our manuscript. We have clarified this in the revised draft.
>
> ---
> **3. Comparison of our dLMs with AR models**
>
> Table 1 of our manuscript compares different attention patterns with only 50B tokens of training, while Table 3 presents results with longer training and additional techniques included. To provide a clearer picture of the comparison between dLMs and AR models, we performed longer continuous pretraining for 300B tokens after submission and report results for the 1.5B/4B/8B Efficient-DLM models, converted from Qwen2.5-1.5B, Qwen3-4B, and Qwen3-8B, respectively.
>
> | Model | Token-per-Forward | Throughput (tok/sec) | Coding (%) | Math (%) | MMLU (%) | Commonsense Reasoning (%) | Avg (%) |
> |:---:|:---:|:---:|:---:|:---:|:---:|:---:|:---:|
> | Dream-7B | 1.00 | 28.11 | 58.92 | 58.39 | 67.00 | 72.83 | 65.30 |
> | LLaDA-8B | 1.00 | 25.04 | 38.10 | 49.13 | 65.86 | 68.50 | 54.92 |
> | Qwen2.5-1.5B | 1.00 | 73.03 | 42.17 | 46.98 | 60.96 | 66.00 | 54.47 |
> | Efficient-DLM-1.5B (Ours) | 1.00 | 68.52 | 42.33 | 42.60 | 57.63 | 62.58 | 52.09 |
> |  | 2.33 | 158.89 | 42.33 | 42.28 | 57.63 | 62.58 | 52.04 |
> |  | 2.69 | 184.48 | 41.79 | 41.80 | 57.63 | 62.58 | 51.77 |
> | Qwen3-4B | 1.00 | 47.13 | 63.85 | 66.27 | 73.19 | 70.91 | 67.97 |
> | Efficient-DLM-4B (Ours) | 1.00 | 44.13 | 62.08 | 67.98 | 71.80 | 70.87 | 67.54 |
> |  | 2.52 | 119.33 | 61.37 | 68.56 | 71.80 | 70.87 | 67.39 |
> |  | 3.01 | 130.24 | 60.96 | 68.10 | 71.80 | 70.87 | 67.18 |
> | Qwen3-8B | 1.00 | 42.51 | 68.45 | 69.84 | 76.93 | 73.71 | 71.58 |
> | Efficient-DLM-8B (Ours) | 1.00 | 40.00 | 67.36 | 69.22 | 77.22 | 74.88 | 71.62 |
> |  | 2.57 | 103.89 | 65.64 | 68.52 | 77.22 | 74.88 | 70.93 |
> |  | 3.10 | 126.43 | 64.95 | 68.21 | 77.22 | 74.88 | 70.65 |
>
>
> As shown in the table above, we observe the following:
>
> (1) For larger-scale models, our Efficient-DLM can match the accuracy of AR models when decoding one token per step. For example, Efficient-DLM-8B achieves +0.04% average accuracy over Qwen3 8B when decoding one token per forward; more aggressive parallel generation can result in a 2.44x throughput increase with only a 0.6% accuracy drop. Also, our Efficient-DLM-8B outperforms LLaDA-8B and Dream-7B by a notable margin.
>
> (2) A single dLM can achieve adaptive accuracy–efficiency trade-offs and outperform AR model families. For example, our Efficient-DLM-8B achieves +2.7% average accuracy and 2.68x throughput compared to Qwen3 4B.
>
> (3) For small-scale models, there remains an accuracy gap between dLMs and AR models, e.g., Efficient-DLM-1.5B still falls behind Qwen2.5-1.5B. We assume this is because smaller models have more difficulty learning to recover corruptions and leverage richer context compared to larger models. That said, the efficiency benefit remains.
>
> We will emphasize this finding in the final version and hope these observations provide a clearer picture of AR-to-dLM conversion.

---

> ### Author Response · Authors · 2025-11-22
> **Response to Reviewer KCLd (Part 2)**
>
> **4. Missing definition**
>
> Thank you for the good catch. $\alpha_t$ denotes the noise schedule, and in our case, it can be simplified to the noise level $t$. We have clarified this in the revised draft.
>
> ---
> **5. Novelty and contributions**
>
> We would like to humbly clarify our contributions and compare them with prior work as follows.
>
> Overall, our work is among the first (alongside some concurrent efforts) to study effective AR-to-dLM conversion that preserves the accuracy of pretrained AR models. This is achieved by identifying the limitations in attention patterns and objectives (i.e., which tokens to mask and learn) in existing works, and by proposing the principles and methodology for successful AR-to-dLM conversion, which lead to a SOTA dLM family.
>
> ***(1)*** For attention patterns, our work identifies the correct attention pattern with a win–win in accuracy and efficiency through a comprehensive study, and simplifies unnecessary designs in prior works.
>
> **Compared to DiffuLLaMa:** Although DiffuLLaMa also studied AR-to-dLM conversion, their adopted bidirectional attention is a suboptimal design and leads to a significant accuracy drop according to Table 1 of their paper and Table 1 of our manuscript. The key novel finding we provide is that maintaining pretrained AR weight distributions with block-wise causality is critical for effective AR-to-dLM conversion.
>
> In addition, DiffuLLaMa adopts token shifts (next-token prediction) to narrow the gap between AR and dLM, and we find that this extra design is unnecessary and simplify it.
>
> By the way, we have cited DiffuLLaMa in Section 2.2 of our submitted manuscript (see “Scaling diffusion language models via adaptation from autoregressive models” in the reference list). We have also added another sentence (highlighted in blue) in the related work section of the revised draft to further acknowledge this prior work.
>
> **Compared to Block Diffusion:** Although Block Diffusion studied block-wise attention patterns, their primary aim is to enable caching. In contrast, we are the first to identify the benefits of block-wise attention in preserving causality for AR-to-dLM conversion, which leads to a +9% average accuracy improvement according to Table 1 of our manuscript. Thus, we propose a strong position for the community: block-wise attention (with clean context) is the way to go, achieving a win–win in accuracy and efficiency for AR-to-dLM conversion.
>
> ***(2)*** In addition, we also propose a new position-dependent token masking scheme in Section 3 of our manuscript. We note that previous works study only the training–test gap in attention patterns of dLMs, while we are the first to pinpoint the training–test gap in token masking schemes and to verify the effectiveness of more strategic token masking schemes. Beyond the proposed solution, this finding and the identified problem may be even more valuable in inspiring future AR-to-dLM training schemes.
>
> ***(3)*** We scale up our techniques to 1.5B/4B/8B dLMs and deliver the Efficient-DLM model family, which outperforms previous dLMs by a notable margin. More importantly, in addition to final model performance, we perform an extensive study on the impact of different design factors to shed light on the underlying principles (summarized in the key takeaways at the end of each section). We also demonstrate the promise of dLMs in one-for-all flexibility for adaptive accuracy–efficiency trade-offs with a single model, as well as their benefits in stronger text embeddings. This is intended to inspire the community regarding additional advantages of dLMs over AR models.

---

> ### Author Response · Authors · 2025-11-22
> **Response to Reviewer KCLd (Part 3)**
>
> **6. Text embedding comparison with LLM2Vec**
>
> We humbly clarify that the main purpose of Section 5.3 of our manuscript is to demonstrate the advantages of dLMs over AR models in providing better text embeddings. This is intended to remind the community about the additional benefits of dLMs over AR models, rather than to claim that diffusion training is the best way to deliver text embedding models.
>
> That said, following your suggestion, we have added a comparison of our latest Efficient-DLM-1.5B/4B models with LLM2Vec by applying their official GitHub code, switching the model to Qwen2.5-1.5B and Qwen3-4B, and performing finetuning using their default dataset and training recipe. The results are shown in the table below, where our Efficient-DLM-1.5B/4B can still achieve better results.
>
> We also want to emphasize that this does not indicate that our dLM is a better scheme for delivering text embedding models than LLM2Vec, since the training scheme and datasets are not exactly the same. Given more time, we can perform a deeper study if you are interested in this comparison. The current comparison is only intended to show that dLMs have reasonably strong text embedding capabilities and have the potential to achieve both good text embedding and good generation simultaneously.
>
> | Model | Retrieval (3) | Ranking (2) | Clustering (3) | Pair Classification (1) | Classification (3) | STS (3) | Average (15) |
> |:---:|:---:|:---:|:---:|:---:|:---:|:---:|:---:|
> | Official Qwen2.5-1.5B  | 20.69 | 40.01 | 21.42 | 24.59 | 32.92 | 49.74 | 31.56 |
> | Qwen2.5-1.5B-LLM2Vec | 21.96 | 42.37 | 21.64 | 45.88 | 49.83 | 53.23 | 39.15 |
> | Efficient-DLM-1.5B (Ours) | 20.51 | 43.62 | 23.95 | 56.28 | 40.68 | 61.52 | 41.09 |
> | Official Qwen3-4B | 19.46 | 39.90 | 21.77 | 33.94 | 29.12 | 50.44 | 32.44 |
> | Qwen3-4B-LLM2Vec | 25.75 | 40.96 | 21.62 | 56.77 | 47.58 | 53.43 | 41.02 |
> | Efficient-DLM-4B (Ours) | 19.99 | 44.54 | 23.81 | 72.02 | 44.72 | 59.97 | 44.18 |
>
> *Note: (number) indicates the number of tasks in this category. For the Classification and STS tasks in this table, we follow LLM2Vec by averaging over English tasks only. As a result, the reported values for Qwen2.5-1.5B and Qwen3-4B on these two tasks differ from those in our manuscript.*
>
> ---
>
> We believe that we have answered all your questions and concerns. Please let us know if you have any follow-up questions!

---

> > ### Author Response · Authors · 2025-11-26
> >
> > Dear Reviewer KCLd,
> >
> > As we approach the end of the discussion period, we would greatly appreciate any further feedback you may have on our responses. We believe we have clarified and addressed all of your concerns and questions, and we welcome any additional comments or suggestions you might have. We are happy to continue the discussion.

---

> > > ### Author Response · Authors · 2025-11-29
> > >
> > > Dear Reviewer KCLd,
> > >
> > > We believe that your major concerns, rating, and comments regarding our use of an “impossible loss” stem from misunderstandings of our method. As clarified in our response, we use exactly the same loss described in the draft and it is certainly a valid and possible loss.
> > >
> > > In addition, your comments on mask counts and the comparison with AR models (for which you referenced an ablation table instead of our final results table) also appear to arise from misunderstandings of our manuscript, and we have clarified all of these points.
> > >
> > > We believe our responses have thoroughly addressed your concerns. We would greatly appreciate it if you could leave a brief comment indicating whether our clarifications were helpful, especially given that the discussion portal will close soon and we have not yet received your feedback. Thank you.

---

### Official Review · Reviewer_xbsE · 2025-11-04

**Soundness:** 3
**Presentation:** 4
**Contribution:** 4
**Rating:** 8
**Confidence:** 3

**Summary:**

The paper studies how to turn autoregressive (AR) models to diffusion language models (dLMs) that are both fast and accurate. The authors identify key ingredients in turning AR models into diffusion language models: block attention to preserve local bidirectional modeling, appropriate block sizes, sampling preferences leaning toward left-to-right, conditioning masking on clean past data, and avoiding token shift. Together, these changes yields efficient and performative dLMs, with upto 2x throughput while also improving accuracy.

**Strengths:**

- The paper tackles an interesting question of distilling AR models into dLMs.
- Many choices in this conversion are elicited and evaluated.
- The resulting method of continued pertaining with block-wise bidirectional attention with clean past context provides significant improvements in efficiency because it allows using many existing methods to speed up inference like the KV-cache.
- Adaptively efficient decoding by changing the confidence threshold in decoding tokens shows near Pareto-optimal trade-offs.

**Weaknesses:**

See questions.

**Questions:**

- what is the role of confidence threshold in decoding for different choices of the half-life? It seems that forcing models to decode later tokens could worsen the trade-off.
- Is the set of "good choices' you discovered unique to pre-trained AR models? It seems true because you made choices to not let the weight go too far. What would happen if I trained with your procedure? That seems like an important comparison, showing whether language priors inside the AR model help learn diffusion models or whether it's really just the proposed set of choices.
- Is there any repeated training? In that Qwen models train on the same data you do?

---

> ### Author Response · Authors · 2025-11-22
> **Response to Reviewer xbsE (Part 1)**
>
> Thank you for acknowledging the novelty of our work and the thoroughness of our experiments, and for your constructive comments! We have addressed all of your questions below.
>
> ---
>
> **1. Relationship between confidence threshold and half-life ratios in position-dependent masking**
>
> We clarify that position-dependent masking, which is controlled by the half-life ratios, is only applied during the continuous pretraining process, without changing the inference-time strategy. At inference time, we still follow the same strategy as fast-dLLM: only confidence-based sampling, controlled by the confidence thresholds, is applied.
>
> Specifically, the goal of position-dependent masking is to mimic test-time behaviors during training, thereby alleviating the training–test-time gap. This is motivated by the observed left-to-right tendency of confidence-based sampling at test time, as analyzed in Section 3.1 of our manuscript, where mask tokens are more likely to be retained in later positions of a block or sentence. Our position-dependent masking mimics this confidence-based sampling during training by assigning higher masking probabilities to later tokens, without changing the inference-time strategy. Thus, given that our method still uses confidence-based sampling at test time, the trade-off will not be worsened.
>
> ---
>
> **2. Whether the identified design choices are good for AR-to-dLM conversion or general dLM training**
>
> This is a very good question! In my understanding, your question is whether the identified good design choices are specific to the AR-to-dLM conversion setting or apply to general dLM training from scratch.
>
> Following your suggestion, we extend Table 1 of our manuscript, which compares different attention pattern choices of dLMs, to a train-from-scratch setting. Specifically, we train three dLMs (same architecture as Qwen3 4B) from scratch with 200B tokens, each with one of the following settings: bidirectional attention, block-wise attention, or block-wise attention with clean context. The results are shown in the table below.
>
> | **Attention Pattern** | **HumanEval** | **HumanEval_Plus** | **MBPP** | **MBPP_Plus** | **GSM8K** | **Avg** |
> |:---:|:---:|:---:|:---:|:---:|:---:|:---:|
> | Bidirectional | 30.49 | 26.22 | 30.60 | 44.71 | 43.29 | 35.06 |
> | Block-wise w/o clean context | 32.93 | 29.88 | **37.20** | 50.00 | 50.57 | 40.11 |
> | Block-wise w/ clean context | **42.68** | **36.59** | 36.80 | **53.44** | **63.00** | **46.50** |
>
> We can observe that:
>
> (1) Block-wise attention still further improves over bidirectional attention under a training-from-scratch setting. We assume this is because block-wise causality favors the autoregressive nature of language and eases learning compared to fully bidirectional training.
>
> (2) Block-wise attention with clean context can notably improve the average accuracy over block-wise attention without clean context. This is consistent with the improvements of conditioning on clean context in the AR-to-dLM setting (Table 1 of our manuscript), which we assume are due to mitigated training–test gaps of dLMs analyzed on page 4 of our manuscript (“The impact of clean context”).
>
> As such, our key insight is that design choices that leverage the autoregressive nature of language and/or help mitigate dLM training–test gaps can be generally beneficial. We believe this set of comparisons is meaningful, and we will extend this discussion to additional design choices in the final version.

---

> ### Author Response · Authors · 2025-11-22
> **Response to Reviewer xbsE (Part 2)**
>
> **3. More information about AR Qwen model baselines**
>
> For the AR Qwen model baselines reported in our manuscript, we measure the results of the official Qwen models on HuggingFace w/o further finetuning. We adopted this setting because our goal is to study whether AR-to-dLM continuous pretraining can recover the accuracy of pretrained AR models while enjoying efficiency benefits. Since the datasets used to train the Qwen models are unknown, we trained our dLMs using the different datasets mentioned in Section 2.2 of our manuscript.
>
> In addition, we have also finetuned the AR Qwen3 4B using an AR loss on the same dataset we used, for 50B tokens, to study the impact of our datasets. We provide the results below.
>
> | **Model** | **HumanEval** | **HumanEval Plus** | **MBPP** | **MBPP Plus** | **GSM8K** | **Minerva Math** | **MMLU** | **ARC-E** | **ARC-C** | **Hellaswag** | **PIQA** | **Winogrande** | **Avg** |
> |:---:|:---:|:---:|:---:|:---:|:---:|:---:|:---:|:---:|:---:|:---:|:---:|:---:|:---:|
> | Qwen3 4B (Official) | 57.32 | 50.61 | **66.80** | **80.69** | **85.44** | 47.10 | 73.19 | 79.12 | 51.62 | **73.66** | 78.07 | 72.06 | 67.97 |
> | Qwen3 4B (Finetuned) | **59.76** | **54.88** | 63.80 | 77.87 | 82.56 | **48.14** | **74.01** | **83.04** | **57.25** | 72.29 | **78.84** | **72.61** | **68.75** |
>
> In general, the finetuned model shows mixed results: in the coding domain, it improves HumanEval but degrades MBPP; in the math domain, it improves Minerva Math but degrades GSM8K, with an overall average accuracy improvement of 0.78%. This indicates that Qwen models are sufficiently pretrained on high-quality datasets, and finetuning on our data can improve accuracy on certain tasks, but only to a limited extent.
>
> ---
>
> We believe that we have answered all your questions. Please let us know if you have any follow-up questions!

---

### Meta-Review · Area_Chair_rb1r · 2025-12-23

**Summary:**

This paper presents Efficient-DLM, a family of diffusion language models obtained by converting pretrained autoregressive models through block-wise attention with context conditioning and position-dependent token masking. The work claims to achieve improved throughput with preserved accuracy, delivering models at 1.5B, 4B, and 8B scales. Reviewers acknowledged the comprehensive empirical studies and thorough experimental evaluation across multiple design factors. The delivered models demonstrate competitive performance compared to existing diffusion language models like Dream and LLaDA. However, significant concerns remain regarding novelty and methodology. The core weakness identified by multiple reviewers is that the approach appears to be a straightforward combination of existing techniques: block-diffusion attention from prior work, autoregressive initialization from DiffuLLaMa, and KV-cache acceleration. While the authors conduct extensive ablations, most observations are considered well-known within the community. Additionally, Reviewer KCLd raised serious concerns about the methodology description, particularly regarding Equation 1 and its implementation, which required extensive clarification. The decision to reject is primarily based on insufficient novelty beyond engineering contributions and the combination of negative scores from two reviewers without compelling evidence that fundamental concerns were resolved.

**Reviewer Concerns:**

The authors provided detailed responses addressing all reviewer concerns with additional experiments including new 8B model results and extended ablations. They clarified that their implementation exactly matches Equation 1 using specialized attention masks, addressed mask count computations, and provided extensive comparisons showing larger models can match autoregressive baselines. The rebuttal emphasized their contribution in identifying block-wise causality as critical for effective conversion, distinguishing their work from prior approaches that used suboptimal bidirectional attention. For Reviewer xbsE, the authors addressed questions about confidence thresholds, training from scratch experiments, and dataset details. For Reviewer mabx, they provided throughput analysis across different context lengths and committed to improving paper organization. For Reviewer at2L, they provided component-wise ablation tables and clarified novelty claims. While the authors claim Reviewer KCLd's concerns stem entirely from misunderstandings, even suppose the reviewer raised the score, the novelty concerns raised by both KCLd and at2L regarding the combination of existing techniques remain fundamentally unresolved regardless of clarifications about implementation details. I personally like the token-dependent masking highlighted by the author, however, in my humble opinion, this training will introduce more autoregressive behaviors to the model, and might not be the ultimate solution, but more like a patch for now. I think this is a good paper regarding engineering success and impact but from the methodology side the contribution is a little bit weak.

**Reviewer Scores:**

Reviewer xbsE would likely maintain their score of 8 given their positive assessment and the authors addressing their technical questions satisfactorily. Reviewer mabx at score 6 might have kept the score. Reviewer at2L at score 4 raised fundamental novelty concerns that the rebuttal attempted to address but likely would have remained at 4, as the core concern about combining existing techniques was not fully resolved. Reviewer KCLd at score 2 with confidence 5 had both methodology concerns and novelty objections. While the methodology clarifications were extensive, the combination of novelty concerns and the strong initial confidence level suggests they would have remained at 2 or possibly moved to 4 at most.

---

### Decision · Program_Chairs · 2026-01-26

Reject